# The Diagnostic Accuracy of Colon Capsule Endoscopy in Inflammatory Bowel Disease—A Systematic Review and Meta-Analysis

**DOI:** 10.3390/diagnostics14182056

**Published:** 2024-09-16

**Authors:** Ian Io Lei, Camilla Thorndal, Muhammad Shoaib Manzoor, Nicholas Parsons, Charlie Noble, Cristiana Huhulea, Anastasios Koulaouzidis, Ramesh P. Arasaradnam

**Affiliations:** 1Institute of Precision Diagnostics & Translational Medicine, University Hospital of Coventry and Warwickshire, Clifford Bridge Rd, Coventry CV2 2DX, UK; cristiana.huhulea@uhcw.nhs.uk (C.H.); r.arasaradnam@warwick.ac.uk (R.P.A.); 2Warwick Medical School, University of Warwick, Coventry CV4 7AL, UK; 3Surgical Research Unit, Odense University Hospital, 5700 Svendborg, Denmark; camilla.thorndal.nielsen@rsyd.dk (C.T.); akoulaouzidis@hotmail.com (A.K.); 4Department of Gastroenterology, Sandwell and West Birmingham Hospitals NHS Trust, Hallam St., West Bromwich B71 4HJ, UK; muhammad_shoaib_manzoor@yahoo.com; 5Warwick Clinical Trials Unit, University of Warwick, Coventry CV4 7AL, UK; nick.parsons@warwick.ac.uk; 6Noblesoft, Stamford, Lincs PE9 3DN, UK; c.noble@noblesoft.co.uk; 7Department of Clinical Research, University of Southern Denmark, 5230 Odense, Denmark; 8Department of Gastroenterology, Pomeranian Medical University, 70-204 Szczecin, Poland; 9Department of Surgery, OUH Svendborg Sygehus, 5700 Svendborg, Denmark; 10Leicester Cancer Centre, University of Leicester, Leicester LE1 7RH, UK

**Keywords:** colon capsule endoscopy, panenteric capsule endoscopy, inflammatory bowel disease, non-polypoidal colonic conditions, colonic inflammation, colonic inflammatory conditions

## Abstract

Colon capsule endoscopy (CCE) has regained popularity for lower gastrointestinal investigations since the COVID-19 pandemic. While there have been systematic reviews and meta-analyses on colonic polyp detection using CCE, there is a lack of comprehensive evidence concerning colonic inflammation. Therefore, this systematic review and meta-analysis aimed to assess the diagnostic accuracy of CCE for colonic inflammation, predominantly ulcerative colitis (UC) and Crohn’s disease (CD). **Methods:** We systematically searched electronic databases (EMBASE, MEDLINE, PubMed Central, and Cochrane Library) for studies comparing the diagnostic accuracy between CCE and optical endoscopy as the standard reference. A bivariate random effect model was used for the meta-analysis. **Results:** From 3797 publications, 23 studies involving 1353 patients were included. Nine studies focused on UC, and ten focused on CD. For UC, CCE showed a pooled sensitivity of 92% (95% CI, 88–95%), a specificity of 71% (95% CI, 35–92%), and an AUC of 0.93 (95% CI, 0.89–0.97). For CD, the pooled sensitivity was 92% (95% CI, 89–95%), and the specificity was 88% (95% CI, 84–92%), with an AUC of 0.87 (95% CI, 0.76–0.98). Overall, for inflammatory bowel disease, the pooled sensitivity, specificity, and AUC were 90% (95% CI, 85–93%), 76% (95% CI, 56–90%), and 0.92 (95% CI, 0.94–0.97), respectively. **Conclusions:** Despite the challenges around standardised disease scoring and the lack of histological confirmation, CCE performs well in diagnosing inflammatory bowel disease. It demonstrates high sensitivity in both UC and Crohn’s terminal ileitis and colitis and high specificity in Crohn’s disease. Further studies are needed to evaluate the diagnostic accuracy of other colonic inflammatory conditions.

## 1. Introduction

In recent years, colon capsule endoscopy (CCE) has gained significant popularity as an alternative to colonoscopy and computed tomography colonography (CTC) for lower gastrointestinal (GI) investigations. CCE witnessed widespread adoption in the Scottish, English, and Danish healthcare systems, producing large-scale studies such as ScotCap, NHS England Pilot, and CareForColon [1,2]. Several systematic reviews were published but focused only on polyps and colorectal cancer (CRC) detection using CCE [3,4,5]. Moreover, a systematic review of artificial intelligence (AI) in CCE indicated that all the attention was given to polyp detection [6].

In comparison, other colonic inflammatory conditions such as inflammatory bowel disease (IBD), including ulcerative colitis (UC) and Crohn’s disease (CD), have not been studied to the same extent in CCE. Even though CD has been studied extensively in small-bowel capsule endoscopy (SBCE) in the literature, CCE is radically different from SBCE, especially since CCE has two cameras rather than the one in the SB capsule [7]. Therefore, the diagnostic accuracy of CCE in UC and CD remains unclear, resulting in the absence of relevant recommendations for using CCE for suspected IBD [8]. In the literature, the latest systematic review was conducted by Tamilarasan et al. on the diagnostic accuracy of IBD using “panenteric capsule endoscopy (PCE)—Crohn’s capsule^TM^ (Medtronic, Minneapolis, MN, USA)” rather than the newest model of colon capsule endoscopy—CCE 2 (PillCam Colon 2^TM^, Medtronic, Minneapolis, MN, USA). The Crohn’s capsule is derived from the reprogrammed and software-redesigned CCE2, addressing a limitation of the original CCE1—the inability to capture the entire small bowel fully. Given the similarity between CCE2 and Crohn’s capsule, that systematic review also employed studies including CCE 1 and 2. However, significant emphasis was placed on small-bowel CD, magnetic resonance enterography (MRE), and PCE instead of CCE [9].

As the utilisation of CCE becomes more widespread, various colonic inflammatory conditions have become more apparent. Therefore, it is crucial to understand the diagnostic accuracy of CCE for these conditions. In addition, a non-invasive diagnostic alternative to colonoscopy would also be beneficial, as colonoscopy can cause significant discomfort. This is especially relevant for IBD patients, who often require frequent surveillance colonoscopies. Therefore, this systematic review and meta-analysis will assess the diagnostic accuracy of inflammatory colonic conditions, predominantly UC and terminal ileal (TI) and colonic CD activity, using CCE.

## 2. Methods

The study protocol was designed based on PRISMA-DTA recommendations [10,11]. The primary aim of the review was to evaluate the per-patient diagnostic accuracy in identifying active UC and terminal ileal and colonic CD using CCE compared to ileocolonoscopy (IC). The secondary aims include assessing the pooled correlation of the detection of IBD severity between CCE and colonoscopy and the diagnostic yield of other miscellaneous pathologies such as diverticular disease, telangiectasia, and haemorrhoids.

### 2.1. Eligibility Criteria

The search included all full texts of clinical and prospective trials that evaluated the diagnostic accuracy of CCE in patients with colonic inflammatory pathologies without language restrictions. This included both adult and paediatric studies. A clear comparison of CCE with IC or transanal enteroscopy as a reference standard is required. There was no restriction on the CCE patients’ recruiting criteria in these studies. Conference abstracts were not included due to the high risk of bias [12]. Review articles, systematic reviews, editorials, study protocols, case reports, and small case series or studies involving ≤10 participants were also excluded.

### 2.2. Information Sources

The databases used to identify relevant publications included EMBASE, MEDLINE, PubMed Central, and Cochrane Library. Additional publications were hand-searched using the references of the extracted studies. The electronic search included all studies up to 8 September 2023, but the search was conducted without any additional time limitations. The search comprised MeSH and non-MeSH terms, including IBD, UC, CD, diverticulitis, infective colitis, checkpoint inhibitor colitis, lower GI bleed, telangiectasia, radiation, and microscopic colitis (see Appendix A, Table A1). The search strings used for each database are available in Appendix A. Grey literature, and unpublished studies were not included.

### 2.3. Study Selection

The title and abstract of all the retrieved studies were reviewed by three of the authors (I.L., C.T., and M.S.M.), and all the studies that did not meet the eligibility criteria were excluded. The inclusion criteria used in the subsequent full-text review were as follows:Comparison between CCE (including both CCE1 or CCE2 or Crohn’s capsule only for terminal ileal and colonic findings) and IC as the comparator arm.The interval between CCE and subsequent IC must be within two weeks.Any colonic inflammation in non-polypoidal pathologies.A prospective study with >10 participants.Use detection or diagnosis of these pathologies as the predominant study endpoint. 


The exclusion criteria are predominantly polyps, CRC, and the use of a small-bowel capsule (see Figure 1). Therefore, five diagnostic studies with inadequate data for data synthesis were excluded from the final analysis but are listed for reference within the appendix (see Appendix A, Table A6).

### 2.4. Data Compilation

The final selection of studies was then reviewed, and data were extracted. The details included the type of pathologies, accuracy, assessment score, severity assessment, extent of disease assessment, type of capsule, comparator arm, regimen of bowel preparation such as the use of prokinetic drugs and boosters, study type, sample size, bowel cleansing quality, and CCE procedure completion rate (see Appendix A, Table A2, Table A3 and Table A4).

### 2.5. Risk of Bias

The selected studies underwent a risk-of-bias assessment utilising the Quality Assessment of Diagnostic Accuracy in Systematic Review-2 and Comparative Study (QUADAS-2 and QUADAS-C) as a component of the quality assurance procedure [13,14]. The risk of bias and the applicability were categorised as low, unclear, or high (see Appendix A, Figure A1).

### 2.6. Data Synthesis and Statistical Analysis

The confusion matrices, which provide the number of true positives (TPs), true negatives (TNs), false positives (FPs), and false negatives (FNs), were extracted from the data provided in each study and categorised into ulcerative colitis (UC), Crohn’s disease (CD), and other pathologies. The key quantitative metrics derived from each study were the sensitivity, specificity, positive predictive value (PPV), and negative predictive value (NPV) [15] (see Figure 1). A diagnostic test accuracy meta-analysis was conducted using a bivariate logistic regression model with random effects (bivariate GLMM) [16,17]. This model, based on pairs of TPs, TNs, FPs, and FNs, was used to calculate pooled estimates of the sensitivity, specificity, diagnostic odds ratio (DOR), positive likelihood ratio (PLR), negative likelihood ratio (NLR), and area under the curve (AUC) using a summary receiver operating characteristic (sROC) curve. The PLR indicates the likelihood that a patient has IBD given a positive CCE result. At the same time, the NLR reflects the likelihood that a patient with a negative CCE result actually has the condition. The AUC serves as a global measure of test performance, with diagnostic accuracy classified as low (AUC < 0.7), moderate (0.7 ≤ AUC < 0.9), or high (AUC ≥ 0.9) [18]. The heterogeneity variance of the logit-transformed sensitivity and specificity of the model and forest plots were used to visualise and explore potential sources of heterogeneity between studies for both sensitivity and specificity. The meta-analysis was carried out using the “lme4” (version 1.1-35.5) [17], “msm” (version 1.8) [19], “mada” (version 0.5.11) [20], and ”lmtest”(version 1.0.0) [21] R packages within the R software (R Core Team, R Foundation for Statistical Computing, Vienna, Austria, https://www.R-project.org/, 25 August 2024) [22]. The correlation coefficients between CCE and ileocolonoscopy in IBD disease detection and disease activity were also reported as a secondary outcome.

### 2.7. Subgroup Analysis and Sensitivity Analysis

When four or more studies were available for subgroup analyses, a meta-analysis was performed following the methods recommended by the Cochrane Diagnostic Test Accuracy Working Group [23]. For studies focusing on CD, the subgroup analysis was limited to two segments: TI and overall colonic disease. In the subgroup and sensitivity analysis, we incorporated pre-defined subgroups, including the CCE1 and CCE2 capsules, Crohn’s capsule and CCE2, different disease extent (CD terminal ileitis vs. CD colitis), different prokinetics usage (such as domperidone, metoclopramide, or none), and various bowel preparation methods (combination of polyethylene glycol, magnesium citrate, senna, and bisacodyl), completion rate (<70%, 70–90%, and >90%), and cleansing quality (adequacy < 70%, 70–90%, and >90%). Post hoc subgroup analysis detected heterogeneity and diagnostic yield variations in these pre-defined categories. 

### 2.8. Publication Bias

In order to assess if systematic differences among the relevant selected studies were missed, we adopted an effective sample-size funnel plot together with the associated regression test of asymmetry, which was the Deeks funnel-plot asymmetry test using DOR and effective sample size (ESS) [24].

## 3. Results

### 3.1. Literature Search

A total of 3797 references from four different databases were identified from the literature search (see Figure 1) [25]. CD studies that assessed the diagnostic accuracy of small-bowel CD only or without IC as a comparator were excluded. Five diagnostic studies did not have adequate data for data synthesis, so they were excluded from the analysis. Twenty-three studies were included in the final analysis, and all the relevant data were extracted by I.L. (see Appendix A, Table A2, Table A3, Table A4 and Table A5) [26,27,28,29,30,31,32,33,34,35,36,37,38,39,40,41,42,43].

### 3.2. Study Characteristics

The selected studies were published between 2006 and 2023, reporting a total number of 1353 enrolled patients. The studies were categorised into UC studies (*n* = 458), CD studies (*n* = 665), and other pathology studies (*n* = 230); overall summaries of the study methodology and results are presented in Table A2 and Table A3 (Appendix A). Due to the limited number of studies on other colonic inflammatory conditions, this systematic review predominantly focuses on inflammatory bowel disease. The findings of other non-polypoidal conditions beyond the scope of IBD are summarised in Table A5 (Appendix A).

Seven of the nine UC studies had adequate data for calculating the diagnostic accuracy. In addition, seven studies reported results utilising the correlation coefficient in UC disease activity and severity detection. In CD studies, six out of nine studies had adequate data for calculating the diagnostic accuracy, and seven studies assessed the overall correlation coefficient in Crohn’s disease detection in the colon and terminal ileum. One of the CD studies used transanal double-balloon enteroscopy as a comparator, and the rest used IC as the gold standard. 

For capsule type, the distribution across studies was as follows: CCE1 (*n* = 5), CCE2 (*n* = 12), both CCE1 and CCE2 (*n* = 1), unspecified CCE type (*n* = 1), and PillCam Crohn’s capsule (*n* = 5) (see Appendix A, Table A4). During the data synthesis, three studies [31,34,44] performed sensitivity, specificity, PPV, and NPV calculations from the raw data. 

For the disease activity assessment score in UC studies, the Mayo score (*n* = 4), UCEIS (*n* = 2), Matt’s endoscopic score (*n* = 2), and Rachmilewitz score (*n* = 1) were used. For the CD studies, the Simple Endoscopic Score (SES-CD) (*n* = 7), Lewis score (*n* = 3), CACDAI score (*n* = 1), and Rugeerts score (*n* = 1) were used. 

### 3.3. Bowel Cleansing and Capsule Completion Rates

Bowel cleansing is pivotal for CCE as it affects diagnostic accuracy. Detailed bowel preparation protocols were reported across all studies, where polyethene glycol (PEG)-based preparation was universally used. Five studies also incorporated additional laxatives as part of their bowel preparation regimen, with notable variations among them. However, four studies did not include the doses. Evaluating whether bowel preparation scores were consistently reported was performed using a 4-point scoring system, including poor, fair, good, and excellent. Only one study did not provide any cleansing scores. The adequacy of overall bowel preparation, defined as fair or above, revealed varying percentages across different studies, ranging from 49% to 98.5% [45].

Another critical performance determinant was the completion rate, which ranged from 68% to 100%. One study did not report the completion rate (see Appendix A, Table A4). 

### 3.4. Risk-of-Bias and Publication Bias Assessment

The studies’ risk-of-bias assessments based on the QUADAS 2 and QUADAS-C tools are presented in Appendix A (Figure A1). Three studies were classified as at high risk of bias, while eleven were identified with an unclear risk of bias. The risk of bias in these studies mainly originated from inadequate specification of the patient selection criteria or unblinding during the selection process, the degree of blinding in the endoscopists and CCE readers, the prolonged period between the index and reference tests, and unclear allocation sequences. Deeks’ regression test showed a *p*-value = 0.23, which suggested no evidence of publication bias (see Appendix A, Figure A2).

### 3.5. Diagnostic Accuracy

Table 1 presents the overall and subgroup analyses of CCE’s diagnostic test accuracy (DTA) in detecting IBD. Table 2 details the heterogeneity variance (τ^2^) and relative sensitivity and specificity as part of the heterogeneity assessment in conjunction with the subgroup analysis. Figure 2 illustrates the bivariate meta-analysis’s forest plot, highlighting CCE’s sensitivity and specificity in IBD. Figure 3 shows CCE’s summary receiver operating characteristic (sROC) curves in diagnosing IBD, UC, and CD.

The overall sensitivity and specificity of CCE in detecting IBD (both ulcerative colitis (UC) and Crohn’s disease (CD)) were 90% (95% CI, 85–93%) and 76% (95% CI, 56–90%), respectively. The sensitivity demonstrated low heterogeneity, with a τ^2^ of 0.13, while the specificity showed high heterogeneity, with a τ^2^ of 1.21. This suggests notable variability in specificity across the studies. An overall area under the curve (AUC) of 0.92 (95% CI, 0.94–0.97) indicates high diagnostic accuracy of CCE for IBD (see Figure 3).

In the subgroup analysis of UC (seven out of nine included studies), CCE’s pooled sensitivity was 92% (95% CI, 88–95%), with low heterogeneity (τ^2^ = 0.041). The pooled specificity, however, was 71% (95% CI, 35–92%), with substantial heterogeneity (τ^2^ = 3.56) and a wide confidence interval. The correlation coefficient between CCE and colonoscopy ranged from 0.75 to 0.86 across the six UC studies. Two studies contributed significantly to the heterogeneity in UC specificity, likely due to their small sample sizes (*n* < 30).

For CD (five out of nine included studies), the pooled per-patient sensitivity of CCE was 90% (95% CI, 85–93%), with a τ^2^ of 0.24, and the pooled specificity was 88% (95% CI, 83–91%), showing low heterogeneity (τ^2^ = 0.13). The AUC for CD detection was 0.87 (95% CI, 0.76–0.98), indicating a moderate diagnostic accuracy of CCE for CD compared with colonoscopy. The correlation coefficient between CCE and colonoscopy in CD studies ranged from 0.49 to 0.82. Further subgroup analysis comparing CD colitis with CD terminal ileitis revealed a sensitivity of 85% (95% CI, 77–90%) and a specificity of 90% (95% CI, 86–94%) for CD colitis compared with 95% (95% CI, 90–97%) and 84% (95% CI, 76–89%) for CD terminal ileitis. Both subgroups exhibited low heterogeneity (τ^2^), suggesting consistency in the results. The relative sensitivity of CCE in detecting CD colitis versus CD terminal ileitis was 0.89 (95% CI, 0.82–0.97), which was statistically significant, indicating its greater sensitivity in detecting terminal ileitis. The relative specificity, however, was not significant (1.08; 95% CI, 0.89–1.18), and the AUC for terminal ileitis (0.95) was higher than for colitis (0.86). The small number of patients and studies involving CD colitis and terminal ileitis limited the accuracy of the AUC 95% CI in the generalised linear mixed model (GLMM), as shown in Table 1.

Comparing UC and CD using CCE, the relative sensitivity was 1.02 (95% CI, 0.96–1.09), and the relative specificity was 0.81 (95% CI, 0.52–1.26), suggesting that CCE has better specificity for CD, although this was not statistically significant. Subgroup analysis of CCE1 versus CCE2 among the UC studies (as no CD study used CCE1) indicated a sensitivity of 89% (95% CI, 82–94%) and a specificity of 46% (95% CI, 44–94%) for CCE1 compared with 95% (95% CI, 86–98%) and 93% (95% CI, 17–99%) for CCE2. Although the relative specificity of CCE1 compared with CCE2 was 0.49 (95% CI, 0.1–2.4), technological advancements likely contributed to the improved accuracy observed with CCE2 despite the difference not reaching statistical significance.

When comparing CCE2 to Crohn’s disease capsules, the subgroup difference in relative sensitivity (0.95; 95% CI, 0.86–1.04) and relative specificity (1.07; 95% CI, 0.96–1.18) showed no statistical significance, suggesting comparable accuracy in IBD detection in both the terminal ileum and the colon. Additional subgroup analyses on completion rates, the use of prokinetics, and bowel preparation regimens revealed no statistically significant differences. This is mainly due to the considerable variability in bowel preparation regimens and booster protocols. Consequently, this emphasises an area deserving of more future research.

## 4. Discussion

Despite the increased acceptance of CCE, the lack of data has made it difficult to position it as a routine lower GI investigation modality globally. Therefore, this current study is an updated systematic review of CCE and the first meta-analysis on CCE in inflammatory bowel disease and other non-polypoidal colonic inflammatory conditions [9].

Based on our meta-analysis, CCE demonstrates a diagnostic accuracy comparable to colonoscopy, with a pooled sensitivity of 90% in detecting IBD overall, 92% in UC disease, and 90% in CD disease activity. The overall AUC was 0.92 in IBD, 0.93 in UC, and 0.87 in CD. These aligned with the findings by Tamilarasan et al. on panenteric capsule endoscopy. It is worth noting that the variance of specificity for IBD overall was 1.21, which is likely attributed to UC’s high variance of specificity of 3.56. This suggests that there was significant heterogeneity in the UC pooled specificity. From analysing the forest plot, the two small studies (Ye et al. [28], *n* = 25; Meister et al. [34], *n* = 13) are the most likely cause of this heterogeneity. After excluding the two small studies in the post hoc subgroup analysis, the heterogeneity variance of UC improved to 2.12 from 3.56, the pooled UC specificity improved to 82% (95% CI, 51–96%) from 71%, and the AUC improved from 0.93 to 0.94. This improvement underscores the importance of adequate sample sizes and power calculations during the planning phase of studies. 

Retrospective power calculations were conducted to evaluate the diagnostic accuracy of CCE, mainly focusing on its pooled sensitivity and specificity relative to colonoscopy. Assuming a true sensitivity and specificity of 0.9 for both UC and CD and using an alpha level of 0.05 with a desired power of 0.9, the calculations (performed using the “pwr” function within the “metafor” package) indicated that each study should ideally have had a minimum sample size of 27, with at least six studies and a total sample size of 162 required to achieve the desired statistical power. Nevertheless, given the limited number of CCE studies on IBD, studies with a minimum sample size of 10 were included in the primary analysis to ensure sufficient studies for an adequate meta-analysis.

In the context of Crohn’s disease, the diagnostic yield of CCE also appeared to be comparable to ileocolonoscopy on the per-patient level, with a low value of heterogeneity. In the subgroup analysis, a lower relative sensitivity of 0.89 was observed when comparing CD colitis to terminal ileitis. This statistically significant finding suggests reduced sensitivity in detecting CD colitis, likely due to the inherent challenges of precise lesion localisation in CCE, particularly around the colonic flexures [48]. While the detection of inflammation posed no significant challenges, difficulty arises in the uncertainty when correlating these findings to those obtained from IC. 

When comparing different capsule iterations, CCE2 showed a significant improvement in specificity compared with CCE1 and a marginal increase in Crohn’s capsule compared with CCE2; there was no statistical difference in the relative sensitivity and specificity when comparing the different models. The improvement could be due to software technology, as described by Tamilarasan et al. and Nia et al. [9,49]. However, the number of studies in these subgroups was very small, indicating the need for more diagnostic accuracy within this area of research. 

Furthermore, while IC stands as the gold standard, its accuracy in detecting small-bowel CD is surpassed by 22% in the diagnostic yield compared with small-bowel CE, as reported in the systematic review conducted by Dionisio et al. [50]. Employing an imperfect gold standard for comparing diagnostic accuracy with CCE, especially in terminal ileitis, raises some concerns. The occurrences of false positives in CCE, especially in the terminal ileum, might indicate cases overlooked by colonoscopy. This introduces a potential source of discrepancy that could diminish CCE’s perceived accuracy despite its potential for a superior diagnostic yield. Therefore, caution must be considered when interpreting the results of diagnostic assessments, particularly in terminal ileal disease. The suggestion by Bruining et al. for addressing this potential challenge was to adopt a panel consensus with the discrepancies. This involves reviewing and discussing the panellists’ endoscopic videos, laboratory results, and clinical notes to confirm the findings and secure the diagnosis. Another included study by Leighton et al. took a distinctive approach by presenting the diagnostic yield of Crohn’s disease without using IC as the gold standard. However, it regarded capsule finding as an endpoint diagnosis, acknowledging the argument above. This led to a 16% improvement in the diagnostic yield of terminal ileitis in CCE compared with IC [46]. However, a drawback of this approach lies in the poor specificity of ileal ulcers, especially in the absence of histological confirmation. This might account for the observed inferior specificity of CCE in Crohn’s terminal ileitis compared with its colonic counterpart.

Furthermore, another critical challenge arises from the uncertainty around the definition of terminal ileum. A recent descriptive study revealed a range of 1 cm to 17 cm of terminal ileum specimens from surgical resections [51]. Another small study also showed that the average length of the examined terminal ileum by colonoscopy was 12.93 ± 6 cm, demonstrating a significant variation in the definition [52]. In clinical practice, the precise measurement of the last 10 cm of the terminal ileum poses a considerable challenge to CCE [46]. This might contribute to the statistically significant increase in CCE’s sensitivity in detecting terminal ileal Crohn’s disease.

Nevertheless, the AUC for the use of CCE in Crohn’s terminal ileitis (0.95) is better than that in Crohn’s colitis (0.86), accepting the potential limitation of the small number of patients and studies. We can only postulate that the reason for the better diagnostic accuracy in the TI might be the more stable capsule’s movement in a smaller lumen and generally better bowel preparation in the terminal ileum compared with its unpredictable rocking motions in the colon and the higher risk of poor colonic bowel preparation [26,46].

Interstudy heterogeneity remains a significant challenge for analysis and data interpretation. This is attributed to different disease activity assessment scores and analysis methods (e.g., per-patient, per-segment, per-lesion, and per-characteristics). Other challenges include the use of diverse correlation coefficients, the uncertainty in securing the diagnosis of Crohn’s disease without an adequate follow-up period, and a decent gold standard reference for comparison. The exclusion of patients with acute severe ulcerative colitis due to the requirement of urgent inpatient investigation and treatment, as well as the deliberate avoidance of individuals with Crohn’s disease with severe stricturing (to mitigate the risk of capsule retention), may introduce a potential selection bias that could inadvertently favour CCE with less-adverse events. In addition, the absence of a pre-registered protocol for this systematic review and meta-analysis might introduce some potential limitations in selective reporting biases and transparency. Ultimately, the inability of CCE to acquire biopsies for further histological assessment remains the primary constraint in IBD assessment, especially in the context of dysplasia detection in UC and the identification of malignancy in this higher-risk patient cohort. However, the increasingly encouraging results of AI in CCE may offer a future solution for detecting dysplasia and subtle malignancies, potentially reducing or eliminating the need for biopsies [27].

During the literature search, the limited number of comparative studies on non-polypoidal inflammation (e.g., infective colitis, checkpoint inhibitor colitis, or diverticulitis) highlighted a significant gap in the research. 

## 5. Conclusions

This systematic review and meta-analysis demonstrated that CCE is comparable to colonoscopy in diagnostic yield in diagnosing UC, Crohn’s terminal ileitis, and colitis in the context of adequate bowel preparation and the procedure completion rate. It has a high sensitivity for both UC and Crohn’s disease, which suggests its possible use as a screening tool, for example, in patients with elevated faecal calprotectin or high risk of IBD. These findings can also guide further appropriate investigations. Regarding UC disease activity assessment, it is unclear whether CCE would have any additional value to faecal calprotectin and colonoscopy, but it could be utilised as an alternative. However, CCE might be useful in assessing the distribution of CD, especially in the terminal ileum where it eludes colonoscopic reach. Further studies are required to evaluate the diagnostic accuracy of other non-polypoidal colonic pathologies.

## Figures and Tables

**Figure 1 diagnostics-14-02056-f001:**
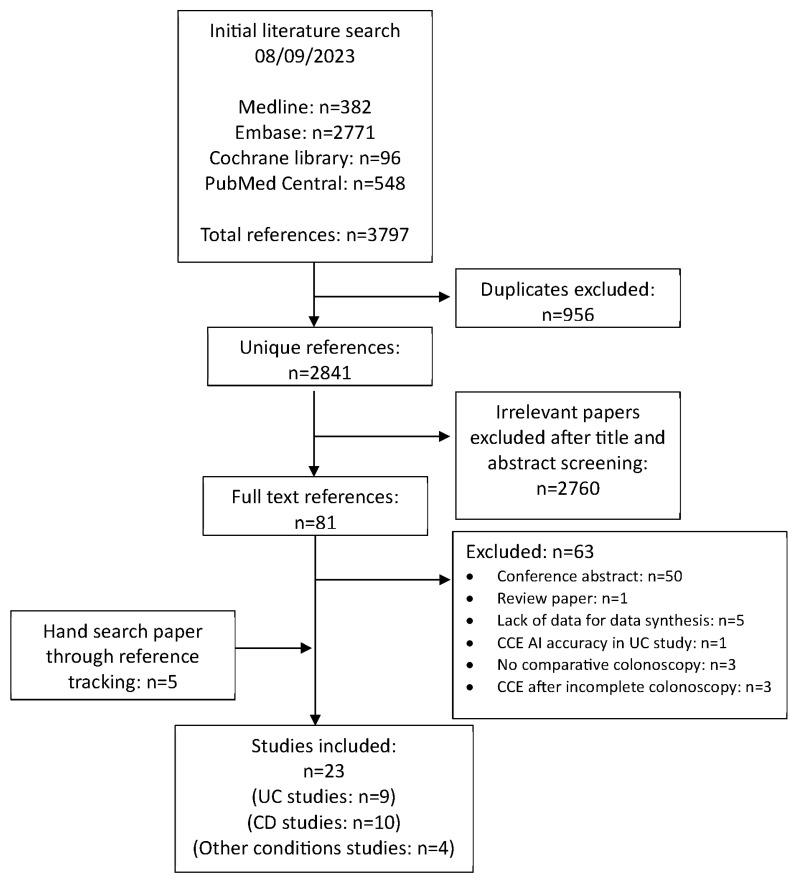
PRISMA flow chart [1].

**Figure 2 diagnostics-14-02056-f002:**
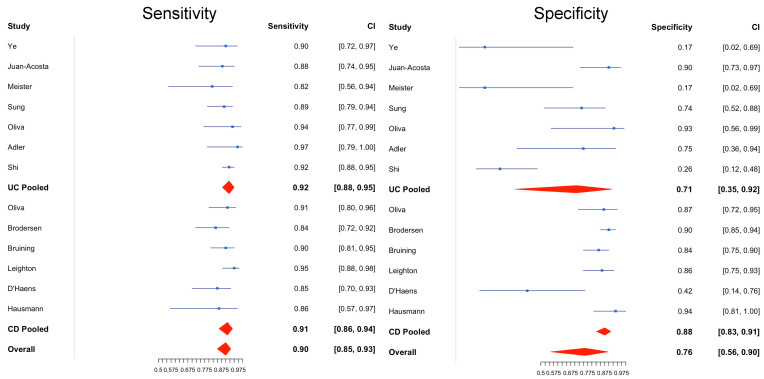
The forest plot of sensitivity (left) and specificity (right) of IBD (including both UC and CD disease activity) detection using CCE. Leighton 2017 [46], Ye C. A. 2013 [28], Juan-Acosta 2014 [29], Shi 2017 [30], Adler 2019 [31], Oliva 2014 [32] Sung 2012 [33], Meister 2013 [34], Hosoe 2013 [35], Hosoe 2018 [36], Oliva 2016 [37], Brodersen 2022 [38], Hausmann 2017 [39], Bruining 2020 [40], Yamada 2021 [41], Papalia 2021 [42], Brodersen 2023 [43], Hall 2015 [47], D’Haens 2015 [44].

**Figure 3 diagnostics-14-02056-f003:**
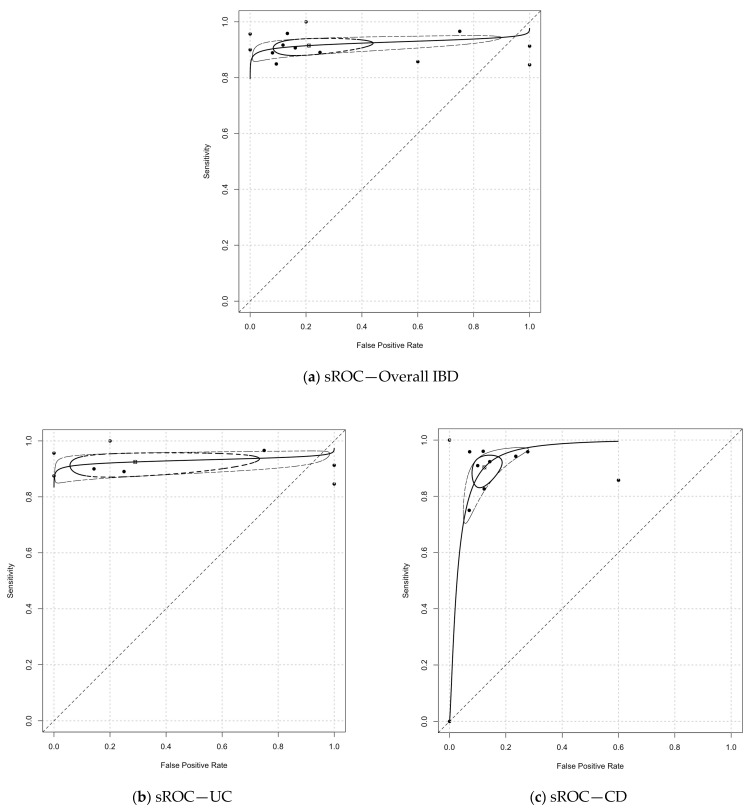
Summary receiver operating characteristic (sROC) curves of CCE for the diagnosis of (**a**) IBD overall, (**b**) ulcerative colitis, and (**c**) Crohn’s disease utilising the generalised linear mixed model (GLMM) from the “glmer” function in the R package “lme4”.

**Table 1 diagnostics-14-02056-t001:** Overall and subgroup analysis for the diagnostic accuracy of CCE in IBD using a generalised linear mixed model (GLMM).

Overall and Subgroup Analysis	Pooled Sensitivity (95% CI)	Pooled Specificity (95% CI)	Pooled PLR (95% CI)	Pooled NLR (95% CI)	Pooled DOR (95% CI)	SROC-AUC (95% CI)
IBD overall (*n* = 13)	0.90 (0.85–0.93)	0.76 (0.56–0.90)	5.43 (5.39–5.46)	0.107 (0.106–0.108)	50.79 (50.27–51.30)	0.92 (0.94–0.97)
No prokinetics (*n* = 3)	0.87 (0.77–0.93)	0.71 (0.27–0.94)	3.04 (−1.07–7.13)	0.18 (0.058–0.31)	16.64 (−14.10–47.37)	NA (*n* < 5)
Metoclopramide (*n* = 4)	0.93 (0.89–0.96)	0.76 (0.58–0.88)	3.86 (1.50–6.22)	0.09 (0.051–0.13)	42.92 (10.88–74.95)	NA (*n* < 5)
Domperidone (*n* = 4)	0.91 (0.84–0.95)	0.93 (0.71–0.99)	13.33 (−7.89–34.56)	0.099 (0.044–0.15)	134.67 (−108.65–377.99)	NA (*n* < 5)
UC pooled (*n* = 7)	0.92 (0.88–0.95)	0.71 (0.35–0.92)	3.19 (−0.24–6.62)	0.11 (0.049–0.16)	30.16 (−14.11–74.43)	0.93 (0.89–0.97)
CCE1 capsule (*n* = 4)	0.89 (0.82–0.93)	0.46 (0.44–0.94)	1.64 (−0.54–3.83)	0.24 (−0.16–0.64)	6.89 (−13.61–27.38)	NA (*n* < 5)
CCE2 capsule (*n* = 4)	0.95 (0.86–0.98)	0.93 (0.17–0.99)	14.54 (−42.6–7.17)	0.052 (0.0078–0.096)	279.62 (−859.73–1419)	NA (*n* < 5)
CD pooled (*n* = 6)	0.90 (0.85–0.93)	0.88 (0.83–0.91)	7.30 (4.90–9.69)	0.11 (0.061–0.16)	65.75 (30.96–100.55)	0.87 (0.76–0.98)
CCE2 capsule (*n* = 4)	0.88 (0.79–0.93)	0.90 (0.84–0.93)	8.74 (5.16–12.31)	0.13 (0.056–0.21)	66.45 (18.34–114.56)	NA (*n* < 5)
Crohn’s capsule (*n* = 2)	0.93 (0.88–0.96)	0.85 (0.76–0.91)	6.07 (3.21–8.92)	0.08 (0.032–0.13)	75.5 (11.45–139.47)	NA (*n* < 5)
CD colitis (*n* = 6)	0.85 (0.77–0.90)	0.90 (0.86–0.94)	8.93 (5.19–12.68)	0.17(0.093–0.24)	54.85 (17.93–89.76)	0.86 (*n* = too small)
CD terminal ileitis (*n* = 5)	0.95 (0.90–0.97)	0.84 (0.76–0.89)	5.89 (3.49–8.29)	0.059 (0.018–0.10)	99.18 (12.95–185.40)	0.95 (*n* = too small)

**Table 2 diagnostics-14-02056-t002:** Relative sensitivity and specificity and subgroup heterogeneity in sensitivity and specificity using generalized linear mixed models (GLMMs).

Overall and Subgroup	Parameter	Logit Scale		Back-Transformed		Heterogeneity Variance τ^2^	Relative Sensitivity and Specificity	Estimate	95% CI
		Mean	Standard Error	Estimate	95% CI				
IBD overall (*n* = 13)	Sensitivity	2.33	0.0037	0.90	0.85–0.93	0.13	NA (only applicable for subroup analysis)	NA	NA
	Specificity	1.60	0.0037	0.76	0.56–0.90	1.21			
No prokinetics (*n* = 3)	Sensitivity	1.90	0.35	0.87	0.77–0.93	0.33	NA (non-dichotomous comparators)	NA	NA
	Specificity	0.91	0.98	0.71	0.27–0.94	0.16			
Metoclopramide (*n* = 4)	Sensitivity	2.613	0.27	0.93	0.89–0.96	0.37	NA (non-dichotomous comparators)	NA	NA
	Specificity	1.146	0.42	0.76	0.58–0.88	0.97			
Domperidone (*n* = 4)	Sensitivity	2.29	0.30	0.91	0.84–0.95	0.00	NA (non-dichotomous comparators)	NA	NA
	Specificity	2.62	0.87	0.93	0.71–0.99	1.39			
UC pooled (*n* = 7)	Sensitivity	2.51	0.25	0.92	0.88–0.95	0.04	UC/CD Relative Sens	1.02	0.96–1.09
	Specificity	0.90	0.78	0.71	0.35–0.92	3.56			
CD pooled (*n* = 6)	Sensitivity	2.23	0.26	0.90	0.85–0.93	0.24	UC/CD Relative Spec	0.81	0.52–1.26
	Specificity	1.96	0.20	0.88	0.83–0.91	0.13			
CCE1 capsule (*n* = 4)	Sensitivity	2.10	0.30	0.89	0.82–0.94	<0.01	CCE1/CCE2 Relative Sens	0.94	0.87–1.01
	Specificity	–0.17	1.48	0.46	0.44–0.94	3.28			
CCE2 capsule (*n* = 4)	Sensitivity	2.97	0.47	0.95	0.86–0.98	0.16	CCE1/CCE2 Relative Spec	0.49	0.10–2.4
	Specificity	2.66	2.15	0.93	0.17–0.99	8.26			
Crohn’s capsule (*n* = 2)	Sensitivity	2.62	0.33	0.93	0.88–0.96	<0.01	CCE2/Crohn’s Cap Relative Sens	0.95	0.86–1.04
	Specificity	1.71	0.28	0.85	0.76–0.91	0.11			
CCE2 capsule (*n* = 4)	Sensitivity	2.01	0.33	0.88	0.79–0.93	0.21	CCE2/Crohn’s Cap Relative Spec	1.07	0.96–1.18
	Specificity	2.19	0.24	0.90	0.84–0.93	0.05			
CD colitis (*n* = 6)	Sensitivity	1.73	0.26	0.85	0.77–0.90	0.05	Colitis/TI Relative Sens	0.89	0.82–0.97 *
	Specificity	2.25	0.24	0.90	0.86–0.94	0.02			
CD terminal ileitis (*n* = 5)	Sensitivity	2.95	0.37	0.95	0.90–0.97	<0.01	Colitis/TI Relative Spec	1.08	0.98–1.18
	Specificity	1.65	0.25	0.84	0.76–0.89	0.03			

* Indicates that the subgroup is statistically significant.

## Data Availability

The raw data supporting the conclusions of this article will be made available by the authors on request.

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
