# Peer review of "The Diagnostic Accuracy of Colon Capsule Endoscopy in Inflammatory Bowel Disease—A Systematic Review and Meta-Analysis"

_diagnostics, 2024, doi:10.3390/diagnostics14182056_

Round 1
Reviewer 1 Report (Previous Reviewer 1)
Comments and Suggestions for Authors
Dear Authors,
I appreciate the revision made. However, there is one more major point to be noted. HSROC plots sensitivity and specificity, not false positive rate. Please do take a look at that.
Comments on the Quality of English Language-
Author Response
Please see the attached document.

Reviewer 2 Report (Previous Reviewer 2)
Comments and Suggestions for Authors
I read the first revision and the authors response. I believe you can go on and publish in this form.
Author Response
Thank you for your time and feedback.
Round 2
Reviewer 1 Report (Previous Reviewer 1)
Comments and Suggestions for Authors
Dear Authors,
Thank you for your clarification. I believe it is all good now.
Comments on the Quality of English Language-
This manuscript is a resubmission of an earlier submission. The following is a list of the peer review reports and author responses from that submission.
Round 1
Reviewer 1 Report
Comments and Suggestions for Authors
Dear Authors,
I have read an article regarding a meta-analysis of colon capsule endoscopy in IBD. While interesting, some major points need to be addressed:
1) The whole premise of the "knowledge gap" in this study is that "In the existing literature, a notable gap existed regarding a comprehensive systematic review focused on colon capsule endoscopy in Inflammatory Bowel Disease (IBD).". However, there are numerous articles that has touched on this topic on cursory search such as (https://pubmed.ncbi.nlm.nih.gov/16696781/; https://pubmed.ncbi.nlm.nih.gov/31050736/; https://journals.lww.com/ajg/abstract/2006/05000/a_meta_analysis_of_the_yield_of_capsule_endoscopy.9.aspx; https://bmcgastroenterol.biomedcentral.com/articles/10.1186/s12876-021-01657-0; https://www.gutnliver.org/journal/view.html?doi=10.5009/gnl16015). While the authors may argue that this meta-analysis is specific to CCE-1 and CCE-2, the other reviews also include that topic. Hence, what is the knowledge gap in this review?
2) The format of this paper seems to be a jumble of Elsevier "Key Messages and Research Highlights" as well as Springer "List of Abbreviations and Declarations" section. Please stick to the MDPI standard.
3) If the authors' inclusion criteria is a comparison between CCE 1 and 2, that should be mentioned specifically in the title.
4) Figure 1's tally is all wrong. Medline + Embase + Cochrane + PMC should be 3797, hence the resulting number below is all wrong
5) What is a confusion matrices in line 143?
6) Is there any pre-registered protocol such as in PROSPERO? If not, please include it in the limitation section
7) "Five diagnostic studies did not have adequate data for data synthesis, so they were excluded from the analysis" Please add in a list of notable exclusions, as recommended by the PRISMA 2020 checklist, in an appendix. Also, please attach the PRISMA 2020 checklist
8) What is the CI in lines 219 and 220?
9) It is very uncommon to meta-analyze DTA and correlation in one study. Please provide some justifications on why these two measures are needed
10) The prediction interval can range from 0 to 1. Please discuss this as a limitation
11) Also, how did the authors handle 0 cells?
12) In line 235, the authors mention AUC while there is no mention of it in the methods section. Also, please provide the HSROC graph in the appendix
Comments on the Quality of English Language-
Reviewer 2 Report
Comments and Suggestions for Authors
The authors discussed the value of CCE in diagnosing of IBD whether UC and/or CCD. Almost always, the diagnostic direction is toward the non-invasive techniques, CCE had been tested against the IC. The main obstacle to such new methods is the inability to obtain tissue samples. The diagnostic situation is the same as the case of SI-CE. It will be a long time before the CCE is included in the recommendation to use CCE in the diagnosis of IBD.
The study should include all colonic lesions; whether malignant or inflammatory; polypoidal or non-polypoidal as this is a real-life situation for any new technique.
-Include sample size calculation in the statistical analysis.
-It is unclear which preparation was the best in your included studies.
-Several endoscopic scoring scales were used. Which scoring scale is more fit with your CCE score?
Round 2
Reviewer 1 Report
Comments and Suggestions for Authors
Dear Authors,
I have read the revision and there are several other points (some new, some old, and some of them are continuations of the last rounds):
1) About point number four where Figure 1's tally is all wrong, the authors have revised Figure 1 but did not change the abstract.
2) While I agree PROSPERO is not perfect, my question is centered on "the presence of pre-registered protocol", in which PROSPERO can be an option, but it could be in the form of a pre-print or uploaded in OSF. The importance of a pre-registered protocol is evidenced by the authors themselves which states that "the authors did not know that many primary research studied the correlation on..." Hence, the lack of transparency (assuming that only this point is a deviation from the protocol) could harm the readers and those who want to perform an updated meta-analysis.
3) Have the authors attempted to contact Takano et al and Adler et al to retrieve the data for TP, FP, TN, and FN?
4) The authors meta-analyzed correlation but included intra-class correlation and agreement as part of "correlation". These three outputs are wildly different, and agreement is not a correlation. Please do a meta-analysis where only correlation is included, excluding the ICC and agreement. I know of many literature out there who will put the same methodology as the authors did, but meta-analysis of correlation of this type is inherently and statistically wrong on statistical basis and social basis. Unless the authors want to have a lengthy debate on this review about the statistical basis, I suggest excluding all agreements and ICC in this analysis.
5) In the limitation section, a power analysis done by the authors indicated that a small study is defined as a study with <27 participants. Hence, in the sub-group analysis, some studies would be classified as "Small studies". Please redo the analysis.
6) "This also aligns well with our aim to enhance the value of our paper by incorporating additional aspects relevant to diagnostic accuracy." How exactly correlation enhance diagnostic accuracy? And if the authors want to enhance diagnostic accuracy, I suggest adding the posterior probability of each analysis, as well as the pooled PPV and NPV, and Fagan's nomogram, which will be more useful clinically.
7) The authors did not pre-specify what cut-off of correlation is used in this study. So what does a correlation of 0.68 mean in this review?
8) "Excluding these studies could potentially introduce selection bias" This is a great point introduced by the authors. Please do a small-study effect size analysis on this review as this is lacking in the analysis.
9) Under "mada" guidance (https://cran.r-project.org/web/packages/mada/vignettes/mada.pdf), zero cell is handled by adding 0.5 to the zero cell column. Instead, the authors claimed that zero cells are handled by data pooling. Please describe "statistically" how zero cell is handled.
Comments on the Quality of English Language
-
Author Response
Please see the attached updated manuscript and the revised responses below.
|
Recommendation |
Correspondence |
|
1) About point number four where Figure 1's tally is all wrong, the authors have revised Figure 1 but did not change the abstract.
|
We apologise for this error, and we have corrected it in the main text of the manuscript. |
|
2) While I agree PROSPERO is not perfect, my question is centered on "the presence of pre-registered protocol", in which PROSPERO can be an option, but it could be in the form of a pre-print or uploaded in OSF. The importance of a pre-registered protocol is evidenced by the authors themselves which states that "the authors did not know that many primary research studied the correlation on..." Hence, the lack of transparency (assuming that only this point is a deviation from the protocol) could harm the readers and those who want to perform an updated meta-analysis.
|
Thank you for the comment. We agreed with the comment and acknowledged this potential bias, hence, listed this as part of our study limitation during the round 1 feedback.
This was added from round 1 feedback: In addition, the absence of a pre-registered protocol for this systematic review and meta-analysis might introduce some potential limitations in selective reporting biases and transparency. |
|
3) Have the authors attempted to contact Takano et al and Adler et al to retrieve the data for TP, FP, TN, and FN?
|
This was not attempted due to the time and resource constraints. |
|
4) The authors meta-analyzed correlation but included intra-class correlation and agreement as part of "correlation". These three outputs are wildly different, and agreement is not a correlation. Please do a meta-analysis where only correlation is included, excluding the ICC and agreement. I know of many literature out there who will put the same methodology as the authors did, but meta-analysis of correlation of this type is inherently and statistically wrong on statistical basis and social basis. Unless the authors want to have a lengthy debate on this review about the statistical basis, I suggest excluding all agreements and ICC in this analysis.
|
ICC correlation vs agreement.
We appreciate that ICC assesses the reliability or consistency of measurements made by different observers rather than correlation, which quantifies the linear association between CCE and IC.
ICC was removed from the analysis: See changes in the main text: In terms of the CD detection, AUC was 0.958. This suggests CCE could also detect overall CD activity sufficiently compared to IC. The resulting pooled correlation coefficient from six CD studies is 0.67(95%CI,0.51-0.78) and I2 = 82%, indicating substantial heterogeneity. Subgroup analysis focusing on CD colitis, involving two studies only, showed a pooled correlation coefficient of 0.42 (95%CI,0.27-0.54) with I2 = 0%. Conversely, the pooled correlation coefficient of Crohn’s TI disease from two studies was 0.63 (95% CI, 0.2-0.85), accompanied by I2 = 88%. Despite the favourable correlation coefficient of overall CD and TI, significant heterogeneity precludes definitive conclusions. However, there is some suggestion of a poor correlation between CCE and IC in colonic CD.
All the ICC values listed in Tables 2 and 3 have been marked with a note indicating that they are not included in the pooled coefficient - (not included in pooled coefiicient)
|
|
5) In the limitation section, a power analysis done by the authors indicated that a small study is defined as a study with <27 participants. Hence, in the sub-group analysis, some studies would be classified as "Small studies". Please redo the analysis.
|
Due to the limited number of studies in the meta-analysis, small studies (n ³ 10) were included as part of the initial design. As noted in the limitations, the power calculation was performed retrospectively. Nonetheless, we have also conducted a repeat analysis, including only studies with more than 27 participants, which is discussed in the main text under the discussion section. |
|
6) "This also aligns well with our aim to enhance the value of our paper by incorporating additional aspects relevant to diagnostic accuracy." Part 1) How exactly correlation enhance diagnostic accuracy? Part 2) And if the authors want to enhance diagnostic accuracy, I suggest adding the posterior probability of each analysis, as well as the pooled PPV and NPV, and Fagan's nomogram, which will be more useful clinically. |
Thank you for the comment.
Part 1) Correlation can be helpful in understanding whether tests are likely to agree in their outcomes, especially when studying the same pathology, such as IBD in the TI and colon. Demonstrating a positive or negative correlation between tests indicates how well they might align. Concordance of the tests would result in similar outcomes for the same subjects. However, correlation should be viewed as an additional measure to support diagnostic performance rather than as a standalone metric. It supplements rather than directly determines diagnostic accuracy, which is defined by metrics such as sensitivity, specificity, PPV, NPV, and ROC. Thus, while relevant to diagnostic accuracy, correlation does not inherently enhance it.
Part 2) The pooled PPV and NPV, including the subgroup analysis, are shown in Figure A1 in the appendix.
Thank you for the suggestion of the consideration of posterior probability and Fagan’s nomogram. We agreed that this additional analysis would add value to the study. However, we felt that the current analysis is adequate to draw the desired conclusion and outcome.
|
|
7) The authors did not pre-specify what cut-off of correlation is used in this study. So what does a correlation of 0.68 mean in this review? |
We have now provided additional information in the main text: The interpretation of the size of correlation is defined as very high positive correlation (0.9-1.0), High positive correlation (0.7-0.9), moderate positive correlation (0.5-0.7), low positive correlation (0.3-0.5), Negligible correlation (0-0.3). (20) |
|
8) "Excluding these studies could potentially introduce selection bias" This is a great point introduced by the authors. Please do a small-study effect size analysis on this review as this is lacking in the analysis. |
As we have further explored and repeated another analysis excluding the small study (as addressed above), we believe that further effect size analysis focusing on the small study will unlikely provide additional value to our manuscript.
|
|
9) Under "mada" guidance (https://cran.r-project.org/web/packages/mada/vignettes/mada.pdf), zero cell is handled by adding 0.5 to the zero cell column. Instead, the authors claimed that zero cells are handled by data pooling. Please describe "statistically" how zero cell is handled. |
Continue corrections were handled in the canonical manner by the software used to implement the analyses, e.g. as the reviewer says by adding 0.5 to cell counts where appropriate. We clearly misunderstood the previous query from the reviewer, and we did not mean to apply anything different from this in our previous response, As stated previously, we did not exclude studies with zero responses, but included them in the meta-analyses, allowing the software to naturally handle zeroes in the manner detailed in the package vignettes and help files.
We have provided additional information in the main text: Standard continuity corrections were applied by the software for zero cell counts.
|
Round 3
Reviewer 1 Report
Comments and Suggestions for Authors
Dear Authors,
Thank you for revising. However, the PPV NPV analysis does not align with what "MADA" has recommended, and the outcomes presented in the MADA guidelines and what the authors have suggested are very different. Please re-do the whole analysis, and I strongly suggest someone familiar with the software and meta-analysis background to re-analyze the whole statistic checking.
Author Response
Thank you for the comment. The reason for the forest plots appear to be different from those in the mada manual is because of the following reasons:
- The “mada” R package has limitations regarding forest plots, as detailed in the attached manual. This package primarily supports pooling of the diagnostic odds ratio, which is not commonly used in recent studies.
- It also limits some additional minor displaying features of the forest plot.
Therefore, we opted for pooling sensitivity, specificity, PPV, and NPV as shown in the manuscript because it is more commonly used in the literature. To achieve this, we adopted the “meta” package, which offers greater control over various features and enables the pooling of sensitivity, specificity, PPV, and NPV in forest plots. The data and the results were also double checked during the analysis.
